# Paradigm Shift: *Candida parapsilosis sensu stricto* as the Most Prevalent *Candida* Species Isolated from Bloodstream Infections with Increasing Azole-Non-Susceptibility Rates: Trends from 2015–2022 Survey

**DOI:** 10.3390/jof9101012

**Published:** 2023-10-13

**Authors:** Iacopo Franconi, Cosmeri Rizzato, Arianna Tavanti, Marco Falcone, Antonella Lupetti

**Affiliations:** 1Department of Translational Research and New Technologies in Medicine and Surgery, University of Pisa, 56126 Pisa, Italy; iacopo.franconi@phd.unipi.it; 2Mycology Unit, Pisa University Hospital, 56126 Pisa, Italy; 3Department of Biology, University of Pisa, 56126 Pisa, Italy; cosmeri.rizzato@unipi.it (C.R.); arianna.tavanti@unipi.it (A.T.); 4Department of Clinical and Experimental Medicine, University of Pisa, 56126 Pisa, Italy; marco.falcone@unipi.it; 5Infectious Diseases, Pisa University Hospital, 56126 Pisa, Italy

**Keywords:** *Candida parapsilosis*, voriconazole, fluconazole, azole-non-susceptibility, candidemia, *Candida* spp. epidemiology

## Abstract

Candidemia is the fourth most common healthcare-related bloodstream infection. In recent years, incidence rates of *Candida parapsilosis* have been on the rise, with differences in prevalence and antifungal susceptibility between countries. The aim of the present study was to evaluate temporal changes in prevalence and antifungal susceptibility of *C. parapsilosis* among other species causing candidemia. All candidemia episodes from January 2015 to August 2022 were evaluated in order to depict time trends in prevalence of *C. parapsilosis sensu stricto* among all *Candida* species recovered from blood cultures as well as fluconazole- and voriconazole-non-susceptibility rates. Secondary analyses evaluated time trends in prevalence and antifungal non-susceptibility according to clinical settings. The overall prevalence of *C. parapsilosis* was observed to increase compared to the prevalence of other *Candida* species over time (*p*-trend = 0.0124). From 2019, the number of *C. parapsilosis sensu stricto* isolates surpassed *C. albicans*, without an increase in incidence rates. Overall rates of fluconazole- and voriconazole-non-susceptible *C. parapsilosis sensu stricto* were both 3/44 (6.8%) in 2015 and were 32/51 (62.7%) and 27/51 (52.9%), respectively, in 2022 (85% cross-non-susceptibility). The risk of detecting fluconazole- or voriconazole-non-susceptibility was found to be higher in *C. parapsilosis* compared to other *Candida* species (odds ratio (OR) = 1.60, 95% CI [1.170, 2.188], *p*-value < 0.0001 and OR = 12.867, 95% CI [6.934, 23.878], *p*-value < 0.0001, respectively). This is the first study to report *C. parapsilosis sensu stricto* as the most prevalent among *Candida* spp. isolated from blood cultures, with worrisome fluconazole- and voriconazole-non-susceptibility rates, unparalleled among European and North American geographical regions.

## 1. Introduction

Candidemia is the fourth most common healthcare-associated bloodstream infection (BSI) [1,2]. Multiple studies over the last two decades have highlighted an overall increase in prevalence and incidence rates in candidemia in all clinical wards [3,4,5]. Patients affected by candidemia are frequently immunocompromised, critically ill, low-weight newborns and subjects who underwent surgical procedures [3,4,5]. Clinical implications and consequences of such an invasive fungal infection include prolonged hospital stay and even death, as candidemia-attributable mortality rates range from 30% to 50% [4,6]. In a recent study conducted by Mete and colleagues, the 30-day overall mortality associated with candidemia was found to be 64% [7].

The distribution and associated epidemiology of *Candida* species among fungal BSI have been extensively studied. Despite differences in the prevalence of non-*albicans Candida* species, which may vary from one geographical region to another, *Candida albicans* is still the most frequently detected fungal pathogen among blood cultures worldwide [5,8]. In the study by Tan et al. [9], published in 2016, on the species distribution and antifungal susceptibility profile of invasive *Candida* spp. bloodstream infections, *C. albicans* was found to be the most frequently isolated species in the Asia–Pacific region, followed by *C. tropicalis* and thirdly *C. parapsilosis,* with prevalence levels of 35.9%, 30.7% and 15.7%, respectively, along with susceptibility rates for fluconazole of 99.7%, 75.8% and 94.8%. In another study, da Matta and colleagues retrospectively investigated the epidemiology of invasive BSIs caused by *Candida* spp. in Central and Latin America [10]. The authors evaluated 40 studies from 2007 to 2016 and found that *C. albicans* was the first species causing candidemia, followed by *C. parapsilosis* and then *C. tropicalis* [10]. Trends in fluconazole-resistant strains showed an increase over time for all the three above-mentioned species, from 0.4% to 1.2% for *C. albicans*, from 0.5% to 2.3% for *C. tropicalis* and from 0% to 2.6% for *C. parapsilosis* [10]. However, defining and illustrating a global picture of clinical trends in the changing epidemiology of invasive candidiasis cannot be so easily achieved as many local hospital-associated factors (i.e., consumption of antifungal agents, patients’ pre-existing conditions, etc.) may alter species distribution. These factors could contribute to explaining the marked differences recovered from one hospital epidemiology to another, even within a single country [2,4,8].

As mentioned above, incidence rates of non-*albicans Candida* species isolated from blood cultures have been on the rise in recent years, with differences in prevalence from one country to another [4,8,11,12]. Data from the SENTRY surveillance program reported by Pfaller and colleagues in 2019 [8] showed that the second most common isolated *Candida* species was *Candida glabrata,* with the exception of Latin America. As previously mentioned, the work by da Matta and colleagues evaluated trends in epidemiology of invasive *Candida* spp bloodstream infections in central and Latin America, highlighting *C. parapsilosis* as the second most frequent fungal isolate causing candidemia [10]. In agreement, data from multiple epidemiological surveys from several centers in Italy [13,14,15], Greece [16], Spain [17] and Belgium [18] as well as outbreak reports from South Africa [19], Latin America [20] and Turkey [7] presented *Candida parapsilosis* as the second most frequently isolated *Candida* species from BSI [21].

*Candida parapsilosis* is a common fungal commensal of the skin flora. Initially, genetic studies on several *C. parapsilosis* strains identified three genotypically different groups defined as “*C. parapsilosis* complex” [22]. Subsequently, these groups were redefined as new species by Tavanti et al. under the names of *C. orthopsilosis*, *C. metapsilosis* and *C. parapsilosis sensu stricto* [23] Furthermore, *C. parapsilosis sensu stricto* has been reported as a causative agent of invasive candidiasis due to its ability to colonize [24] and infect medical devices and intravenous catheters [25], predominantly affecting critically ill patients and neonates [26,27]. Based on its capacity to form biofilm and its virulence factors [28], *C. parapsilosis* has been identified as a healthcare-associated pathogen, spreading via hand carriage and persisting in the hospital environment [26]. In recent years, another peculiar aspect of this microorganism has been pointed out by researchers regarding its antifungal susceptibility [29,30,31]. Indeed, parallel to the reported increase in rates and prevalence of *C. parapsilosis* from BSI among the above-mentioned geographic regions, another clinical phenomenon has been described: decreased susceptibility to echinocandin [32] and azole compounds [29,33]. Azole-resistance is correlated with previous or current exposure to antifungal therapy [34,35]. Voriconazole-resistance can also be developed during and/or after treatment with fluconazole, as cross-resistance among azole compounds is a common phenomenon in *C. parapsilosis* [34]. In a meta-analysis from Yamin and colleagues [36], rates of azole-resistance were listed according to geographical regions, identifying 27.7% fluconazole-resistant isolates in Africa and 13.3% across Europe. Overall voriconazole-resistance was 4.7%, with the highest prevalence reported in South Africa (19.7%) [36]. Data from European surveys differ greatly from one country to another, with the highest rates of azole resistance reported in Italy, of 33% for fluconazole and 18.2% for voriconazole [15], and Turkey, where susceptibility to fluconazole in *C. parapsilosis* accounted for 49% of all isolates collected from 2013 to 2017 [7]. Thankfully, in this scenario, echinocandin resistance is still a limited phenomenon seldomly described, even if reports of resistant isolates along with tolerance and heteroresistance have been reported at increasing rates [30,37].

According to these findings and other reports, the World Health Organization (WHO) has listed *C. parapsilosis* among the 19 groups of fungal pathogens causing human invasive diseases with a high risk of mortality and/or morbidity [38].

In consideration of the reported rise in prevalence of *C. parapsilosis* and the worrisome decrease in antifungal susceptibility, there is an increasing need to obtain novel data on the evolution of local epidemiology, to better define time trends in prevalence and antifungal susceptibility of *C. parapsilosis* affecting critically ill patients.

The present study aimed to provide a deeper insight into the prevalence and antifungal susceptibility trends over time of *C. parapsilosis* among other *Candida* species causing candidemia.

## 2. Materials and Methods

This was a retrospective surveillance study conducted in the Mycology Unit of Pisa University Hospital, a tertiary-level hospital. Data were extracted from a clinical database from 1 January 2015 to 31 August 2022. The authors searched for and extracted data on all *Candida* spp. isolated from BSIs over the study period. For each species and each affected patient, only the first positive blood culture for *Candida* spp. over a 30-day period was included in this study [39]. BSI episodes caused by the same species were, therefore, excluded if detected within 30 days from the first positive blood culture. Age, sex and clinical settings were reported. Clinical settings were classified as internal medicine, surgical and intensive care wards. All isolates recovered from patients of all ages were included in this study. Clinical setting differentiation and categorization for pediatric clinical wards followed the same scheme reported above for adults. Neonatal cases were considered as ICUs. Hematology and oncohematology wards were considered as internal medicine.

In order to describe the clinical context in which this study was performed, according to the latest report provided by the healthcare management in 2021, Pisa University Hospital hosted 1108 beds and had 49,045 admissions in the same year. Surgical wards’ total admissions and beds were 21,279 and 444, respectively. Internal medicine’s total admissions and beds in 2021 were 20,268 and 439, respectively. ICU’s total beds were 108. In addition, a total of 262 surgical transplantation procedures were performed as follows: 140 liver, 81 bone marrow, 33 kidney and 4 pancreas. Data regarding hospital admissions, distribution of beds in relation to medical activities and specialties as well as surgical transplantation procedures were in line with those reported in previous years with the exception of 2020, when a reduction in the number of admissions was observed.

For the entire study period (2015–2022), yeast isolates recovered from positive blood cultures were identified with MALDI-TOF MS (MALDI Biotyper Compass^®^ Bruker Daltonics GmbH & Co. KG, Bremen, Germany). The microbiology laboratory supervised and handled the complete management of blood cultures taken from patients with suspected sepsis. The diagnostic workflow of BSIs in our hospital is as follows: Blood cultures (BACTEC Plus/F Aerobic, BACTEC Plus/F Anaerobic, BACTEC Peds Plus/F) from patients with suspected sepsis are sent to the microbiology laboratory where they are incubated in the BD BACTEC FX system (Becton Dickinson and Company; Milan, Italy) at 35 ± 1.5 °C for up to 7 days. All positive blood cultures then undergo Gram staining and are plated onto Sabouraud agar plates amongst other solid media. Plates are then incubated for 24 h at 35–37 °C. Yeast colonies are then identified using MALDI-TOF MS. A commercial colorimetric method is used to assess antifungal susceptibility. From January 2015 to April 2020, the antifungal susceptibility test used was Sensititre YeastOne Y010© (Thermo Fisher Diagnostics B.V., Lansmeer, The Netherlands). From May 2020 to August 31st 2022, the applied method was Merlin MICRONAUT-AM Antifungal Agents MIC© (Bruker Daltonics GmbH & Co. KG, Germany).

Minimum inhibitory concentration (MIC) values were evaluated using the current CLSI guideline M27Ed4E and EUCAST documents [40,41]. The CLSI methodology and the interpretation of results was according to the reference standard for Sensititre YeastOne Y010©, while EUCAST recommendations and result interpretation were applied with Merlin MICRONAUT-AM Antifungal Agents MIC©. Yeast isolates were cultured in aerobic conditions at 37 °C for 24–48 h, and then multiple isolated colonies of yeast (3 to 5) were suspended in sterile distilled water. The solution was homogenized with the use of a vortex in order to obtain a concentration of 0.5 McFarland detected with a spectrophotometer. This yeast suspension was then resuspended in RPMI 1640 at a final inoculum of 0.5 × 10^5^ and 2.5 × 10^5^ CFU/mL according to the EUCAST reference method when microtitration was performed with Merlin MICRONAUT-AM Antifungal Agents MIC©. The final concentration of the inoculum suspension was 0.5 × 10^3^–2.5 × 10^3^ cells/mL according to the CLSI guidance document when antifungal susceptibility testing was performed using colorimetric broth microtitration with Sensititre YeastOne Y010©. Then, 96-well microdilution plates were incubated at 35–37 °C and checked for growth of the fungal microorganism at 24 h and 48 h. All assays where positive growth control was clear at 24 h were then analyzed and antifungal MIC were interpreted accordingly. If positive growth control was poor at 24 h, the incubation period was extended to 48 h. Comparison between the growth control well and antifungal testing wells was the reference analytical method in order to define MIC values for each antifungal compound. Tested antifungal compounds were fluconazole, voriconazole, posaconazole, micafungin, caspofungin, anidulafungin and amphotericin B. According to hospital protocols, resistant phenotypes under the broth microdilution method were retested for each antifungal molecule presenting an unusual phenotype. The test applied to confirm these results was the E-test strip (Liofilchem^®^ S.r.l., Roseto degli Abruzzi (TE), Italy).

According to the CLSI reference standard document, susceptible *Candida parapsilosis sensu stricto* isolates had fluconazole MIC ≤ 2 mg/L, voriconazole MIC ≤ 0.12 mg/L, anidulafungin ≤ 2 mg/L, caspofungin ≤ 2 mg/L and micafungin ≤ 2 mg/L [40]. For amphotericin B- and posaconazole-susceptibility interpretation, MIC values were compared to epidemiological cutoff values (ECVs). MIC values for *C. parapsilosis* above the ECV (amphotericin B > 1 mg/L, posaconazole > 0.25 mg/L) were retested with the E-test strip. In cases of concordant results, the yeast isolate was classified as “non-wild-type” [42]. According to the EUCAST reference standard document, susceptible *Candida parapsilosis sensu stricto* isolates had fluconazole MIC ≤ 2 mg/L, voriconazole MIC ≤ 0.125 mg/L, posaconazole MIC ≤ 0.06 mg/L, anidulafungin MIC ≤ 4 mg/L, micafungin MIC ≤ 2 mg/L and amphotericin B MIC ≤ 1 mg/L. As reported in the EUCAST document, isolates susceptible to both anidulafungin and micafungin were considered susceptible also to caspofungin [41].

The incidence of candidemia per 10,000 patient-days was calculated for each *Candida* species; total patient-day data per year were obtained from a publicly available repository. Data regarding hospital patient-days and medical activities and reports can be accessed without restriction from the Azienda Ospedaliero-Universitaria Pisana through the following link: https://www.ospedaledipisa.it/index.php?option=com_content&view=article&id=5650:relazione-sanitaria-2021&Itemid=112 (accessed on 30 September 2023) [43] (see Data Availability Statement section).

In all statistical analyses, *C. parapsilosis sensu stricto* isolates with an intermediate or resistant phenotype for fluconazole and/or voriconazole were regrouped as “non-susceptible” for the corresponding molecule.

### Statistical Analysis

Multivariate logistic regression analysis and a score test for the trend of odds (*p*-trend) calculation were performed to evaluate both the increase in overall prevalence of *C. parapsilosis sensu stricto* among all *Candida* species recovered from BSI and the fluconazole-/voriconazole-non-susceptibility rates among *C. parapsilosis sensu stricto* isolated from BSI according to different clinical settings over time. A non-parametric Mann–Kendal trend test was performed to investigate if there was an increase in the incidence of candidemia over time and an overall increase in the prevalence of non-susceptibility to both azole compounds over time. A *p*-value of <0.05 was considered statistically significant. All statistical analysis was performed using STATA 11© (Copyright 1985–2019, StataCorp LLC, Statistics/Data analysis StataCorp, 4905 Lakeway Drive, College Station, TX, USA), and figures were made using Graphpad Prism 9© (GraphPad Software, 2365 Northside Dr. Suite 560, San Diego, CA, USA).

## 3. Results

A total of 1039 candidemia episodes were evaluated. The mean and median age of the study population were 68.9 and 73, respectively, and age of study population ranged from 1 to 115 years. The male population totaled 556 (53.46%) and the female population 483 (46.4%). The overall species distribution from 2015 to 2022 showed 412 (39.7%) cases of *C. albicans*, 371 (35.7%) of *C. parapsilosis sensu stricto*, 147 (14.1%) of *C. glabrata*, 45 (4.3%) of *C. tropicalis* and 64 (6.2%) of others (19 *C. krusei*, 17 *C. guillermondii*, 10 *C. orthopsilosis*, 4 *C. dubliniensis*, 4 *C. metapsilosis*, 3 *C. lusitanie*, 2 *C. incospicua*, 2 *C. pelliculosa*, 1 *C. famata*, 1 *C. rugosa*, 1 *C. lipolytica*). The total number of isolates for each *Candida* species per year is reported in Figure 1. Since the pooled prevalence of *C. orthopsilosis* and *C. metapsilosis* among all candidemia episodes was 1.3%, we decided to perform analysis only for *C. parapsilosis sensu stricto* rather than for the entire complex.

Species distribution and prevalence of each *Candida* species are reported in Table 1.

According to clinical settings, 578 (55.6%) of all candidemia episodes were detected in internal medicine wards, 203 (19.5%) in surgical wards and 258 (24.8%) in intensive care wards across the study period. Incidence rates of overall candidemia and for each *Candida* species are reported in Table 1. The non-parametric Mann–Kendal trend test showed no statistically significant increase in overall incidence of candidemia over time (*p*-value = 0.1331).

The total number of neonatal infections due to *Candida* spp. was surprisingly low, with only three *Candida albicans* bloodstream infections among newborns. A similarly low number of pediatric cases was observed with two bloodstream infections, one caused by *Candida parapsilosis sensu stricto* and the other by *Candida albicans*.

According to the redefinition of susceptible/non-susceptible categories, “non-susceptibility” accounted for 4 *C. parapsilosis sensu stricto* isolates for fluconazole (3%) and 22 *C. parapsilosis* isolates for voriconazole (19.6%). All isolates presenting intermediate and resistant phenotypes to fluconazole and voriconazole were retested with the E-test strip, which confirmed the non-susceptible phenotype.

### 3.1. Increasing Prevalence of Candida parapsilosis in Comparison to Other Candida Species Isolated from Blood Cultures

The prevalence of *C. parapsilosis sensu stricto* among other *Candida* species in 2015 was 44/126 (34.9%). The prevalence of *C. parapsilosis sensu stricto* among other *Candida* species in 2022 was 51/106 (48.1%). Since 2019, the number of *C. parapsilosis sensu stricto* isolates detected from blood cultures reached and surpassed that of *C. albicans* over each year of study: 37.8% vs. 36.3% in 2019, 35.9% for both species in 2020, 39.8% vs. 30.4% in 2021 and 48.1% vs. 34.9% up to August 2022. A score test for the trend of odds (*p*-trend) showed a statistically significant increase in the overall prevalence of *C. parapsilosis sensu stricto* compared to the prevalence of other *Candida* species over time (*p*-trend = 0.0124). The score test for the trend of odds subsequently evaluated time trends in the distribution and prevalence of *C. parapsilosis sensu stricto* in the different clinical settings (Table 2).

The prevalence of *C. parapsilosis sensu stricto* isolated from blood cultures among other *Candida* species statistically increased over the study period in intensive care units from 9/25 (36%) in 2015 to 16/27 (59.2%) in 2022 (*p*-trend < 0.0001). Data regarding surgical wards showed that the prevalence of *C. parapsilosis sensu stricto* isolated from blood cultures among other *Candida* species statistically increased over the study period from 7/24 (29.2%) in 2015 to 10/16 (62.5%) in 2022 (*p*-trend < 0.05). The same statistical analysis reported no significant increase in *C. parapsilosis sensu stricto* in internal medicine wards over the study period (*p*-trend = 0.609). No association was found between age and sex with the increased prevalence of *C. parapsilosis sensu stricto*.

### 3.2. Increasing Fluconazole- and Voriconazole-Non-Susceptibility Rates in Candida parapsilosis in Comparison to Other Candida Species Isolated from Blood Cultures

The overall rates of fluconazole- and voriconazole-non-susceptibility were both 3 (6.8%) out of 44 *C. parapsilosis sensu stricto* in 2015, whereas in 2022, the rates were 32/51 (62.7%) and 27/51 (52.9%), respectively. The numbers of isolates that were non-susceptible to fluconazole per year were 3 (6.9%) out of 43 *C. parapsilosis sensu stricto* in 2015, 2/44 (4.5%) in 2016, 1/41 (2.4%) in 2017 and 8/29 (27%) in 2018. Identical numbers were found for voriconazole, except for 2016 (1/44, 2.2%). From 2019 to 2022, percentages of fluconazole-non-susceptible isolates peaked at 24 out of 51 (47.1%) in 2019, 17/47 (36.2%) in 2020, 45/64 (70.3%) in 2021 and 32/51 (62.7%) up to August 2022. In the same study period, the number of *C. parapsilosis sensu stricto* isolates non-susceptible to voriconazole were 23/51 (45.1%) in 2019, 13/47 (27.7%) in 2020, 36/64 (56.3%) in 2021 and 27/51 (52.9%) in 2022. In 2021 and 2022, the number of candidemia episodes of azole-non-susceptible *C. parapsilosis sensu stricto* recovered in ICUs was higher than the overall number of other *Candida* spp. isolates recovered from blood cultures in the same clinical ward. Distributions of *C. parapsilosis sensu stricto* isolates non-susceptible to fluconazole or voriconazole per year according to clinical settings are reported in Table 3.

The overall cross-non-susceptibility between fluconazole and voriconazole was 85%. All isolates non-susceptible to voriconazole were also non-susceptible to fluconazole. The score test for the trend of odds (*p*-trend) showed a statistically significant increase in overall rates of fluconazole- or voriconazole-non-susceptibility in *C. parapsilosis sensu stricto* over time (*p*-trend < 0.0001 for both antifungals, Figure 2). A non-parametric Mann–Kendall trend test showed and confirmed this statistically significant increase in prevalence of both fluconazole- and voriconazole-non-susceptibility (*p*-value < 0.05 for both antifungals) in *C. parapsilosis sensu stricto*.

Secondary analysis evaluated time trends for rates of fluconazole- or voriconazole-non-susceptibility in *C. parapsilosis sensu stricto* according to different clinical settings. Fluconazole-non-susceptibility in *C. parapsilosis sensu stricto* isolated from blood cultures statistically increased over the study period in all clinical settings: intensive care wards (*p*-trend < 0.0001), surgical wards (*p*-trend < 0.0001) as well as internal medicine wards (*p*-trend < 0.0001). Also, voriconazole-non-susceptibility in *C. parapsilosis sensu stricto* isolated from blood cultures statistically increased over the study period in all clinical settings: intensive care wards (*p*-trend < 0.0001), surgical wards (*p*-trend < 0.0001) as well as internal medicine wards (*p*-trend = 0.0015). Multivariate logistic regression analyses adjusted for year of study and clinical settings confirmed an overall increase in the risk of detecting fluconazole- or voriconazole-non-susceptibility in newly detected *C. parapsilosis sensu stricto* isolates from blood cultures (*p*-value < 0.0001).

The risk of detecting fluconazole-non-susceptibility was found to be higher in *C. parapsilosis sensu stricto* compared to other *Candida* species (OR = 1.600, 95% CI [1.170, 2.188], *p*-value < 0.0001). The risk of detecting voriconazole-non-susceptibility was found to be higher in *C. parapsilosis sensu stricto* compared to other *Candida* species (OR = 12.867, 95% CI [6.934, 23.878], *p*-value < 0.0001).

Interestingly, no resistance to echinocandins was reported across the entire study period for any *C. parapsilosis sensu stricto* isolates.

Age and sex were not found to be associated with a higher risk of detecting non-susceptible isolates and, therefore, were not included as covariates in the analysis. Secondary analysis compared different clinical settings in terms of ward-related risk of detection of fluconazole- or voriconazole-non-susceptible *C. parapsilosis sensu stricto* from blood cultures. The detection of *C. parapsilosis sensu stricto* in intensive care wards was associated with a higher risk of being non-susceptible to fluconazole (OR = 5.547, 95% CI [3.038, 10.126], *p*-value < 0.0001) or voriconazole (OR = 7.684, 95% CI [4.152, 14.221], *p*-value < 0.0001) when compared to internal medicine wards. No significant difference was found either between intensive care wards and surgical wards or between surgical wards and internal medicine wards.

## 4. Discussion

This study has demonstrated an epidemiologic shift in candidemia isolated at Pisa University Hospital, as *Candida parapsilosis sensu stricto* has become the most prevalent *Candida* species isolated from blood cultures, with the highest rates of non-susceptibility to azole compounds among all other *Candida* species.

Looking at the in-hospital distribution of *C. parapsilosis sensu stricto* isolates according to clinical settings, the present data show that the prevalence of *C. parapsilosis sensu stricto* among other candidemia isolates has increased over time in both intensive care and surgical wards, but not in internal medicine wards. Furthermore, trends in fluconazole- and voriconazole-non-susceptibility have registered an increase in all clinical settings. Surprisingly, no statistically significant increase in incidence rates of candidemia were found, thus highlighting that azole-non-susceptible *Candida parapsilosis sensu stricto* has essentially replaced the in-hospital ecologic niche that was previously occupied by fluconazole- and voriconazole-susceptible *Candida parapsilosis sensu stricto* and other *Candida* species. The results reported in the present study show a higher percentage of *C. parapsilosis sensu stricto* among candidemia isolates and higher azole-non-susceptibility rates (cross-non-susceptibility between fluconazole and voriconazole was 85% overall) than data reported by another Italian study [15] and other surveys from Spain [17] Greece [16], Turkey [7], Belgium [18] and Sweden [44]. Herein, we chose to focus on a broad concept of “non-susceptibility” instead of pure resistance as, in clinical practice, the selection of the most effective antifungal therapy in case of invasive fungal infections, as for the case of candidemia, takes into consideration many aspects but first and foremost its full susceptibility observed through in vitro susceptibility tests [6].

Our results should be interpreted with caution. Due to the lack of a molecular method in our hospital for the detection of underlying resistance mechanisms in *Candida* spp., we were not able to perform any adjunctive confirmatory test other than the E-test strip, which still follows a phenotypic approach. Regardless, whole-genome sequencing and adjunctive molecular tests will be the subject of future investigations and research.

A second study limitation lies within the retrospective nature of the work presented. Unfortunately, according to the kind of information stored in the database, the authors were unable to gain more specific information regarding both the source of candidemia and prior exposure to antifungals. Evaluating these parameters along with survival rates and treatment outcomes with prospective studies will certainly be the subject of future research.

Another major drawback of this study that must be outlined is the switch in the antifungal susceptibility testing method during the study period. However, it is also important to point out that a confirmatory E-test strip was still applied in all selected cases of azole-non-susceptibility. In addition, according to results by Pfaller et al. [45], essential and categorical agreements between EUCAST and CLSI broth microdilution methods for antifungal susceptibility testing were high (>90%) for fluconazole, voriconazole and micafungin. Despite this, the MIC values obtained with the EUCAST reference method happened to be lower than those found with CLSI, except for caspofungin [45]. Philips and colleagues [46] compared the performances of the two commercial methods utilized in this study. According to their results, MIC values evaluated with Sensititre were higher than those found with MICRONAUT for all antifungals. These findings are aligned with those previously reported by Pfaller. Interestingly, essential agreement was found to be high (98%) for fluconazole, which is one of the two major antifungal compounds analyzed in this study [46]. On the other hand, statistically significant differences between the two methods were reported with voriconazole. Indeed, higher MIC values were reported when using the Sensititre method, with a higher percentage of *Candida* spp. isolates found to be non-susceptible to this molecule compared to the MICRONAUT method. Therefore, it is undeniable that switching from one method to another is a study limitation; however, this might have only slightly affected fluconazole interpretation due to the high essential agreement between the two methods. As for voriconazole, since overestimation of resistance was reported to be associated with the Sensititre method and the increase in resistance rates in our epidemiological study was observed midstream, we believed that E-test confirmation was a reasonable way to show that the increase in resistance was not merely a function of the test method.

According to international guidelines [6] and in-hospital procedures and protocols, it is not recommended to use other azoles if *Candida* spp. is resistant to fluconazole, but echinocandins.. In this study, since the azole-non-susceptibility detected with in vitro testing also affected voriconazole, we decided to include and report this molecule in all analyses.

This study was undertaken in 2022, when concern was raised by different clinicians within the hospital regarding *C. parapsilosis sensu stricto*-resistant infections. Therefore, the first aim of the study was to retrospectively investigate all *Candida* bloodstream isolates in order to evaluate the real numbers underlying the azole-resistant *Candida parapsilosis sensu stricto* infections. Studying the in-hospital spread of this/these strain/s will of the utmost importance for future research and infection control purposes, as such strain/s might spread among patients within the same room/ward, as reported in a Spanish study conducted by Díaz-García et al. [47]. These authors defined the clinical and molecular epidemiology of a similar strain, where spread of fluconazole-resistant *C. parapsilosis* was observed and investigated across different hospitals in Madrid [47]. However, we were unable to retrieve isolates prior to 2021, when the increase in azole-non-susceptibility took place, or information regarding clusters of patients’ rooms/wards where recovery of azole-non-susceptible *C. parapsilosis sensu stricto* was higher than others. Following these assumptions, this study focused first and foremost on data collection and analysis, and results were reported without a definitive explanation for this intriguing phenomenon. In this regard, finding a single explanation might be hard to accomplish. First, Pisa University Hospital presents intense transplantation surgical activity, as it is a national referral center for conjunct pancreas and kidney transplantation, along with pancreas and kidney alone, and liver transplantation among solid organ transplant procedures. Consequently, there is a high number of patients within the hospital at risk of acquiring an azole-non-susceptible *Candida parapsilosis sensu stricto* infection. Second, our hospital also has an ICU that is a regional referral center for burn patients, thus representing another substrate prone to this specific kind of infection. Indeed, when looking at the data presented, a statistically significant increase in the prevalence of *C. parapsilosis sensu stricto* isolated from blood cultures and its associated azole-non-susceptibility rates was found in both ICUs and surgical wards, confirming their role as in-hospital ecological niches for the spread of azole-non-susceptible *C. parapsilosis sensu stricto*. Third, in consideration of the fact that Pisa University Hospital serves patients coming from all Italian regions, intensifying the connections with patients treated in other healthcare centers with different epidemiological landscapes, the possibilities of inter-hospital transmission of different clades of azole-non-susceptible *C. parapsilosis sensu stricto* are higher than for a non-referral center. Lastly, the SARS-CoV-2 pandemic has greatly influenced local epidemiology of healthcare-associated fungal infections, with a rise in multidrug-resistant *Candida* spp. Indeed, in an interesting study conducted by Ramos-Martinez et al. [48] showed that the COVID-19 pandemic greatly affected the incidence of candidemia, especially in the case of *C. parapsilosis*. The authors highlighted an increase in both incidence rates and associated fluconazole-resistance of *C. parapsilosis* bloodstream infections during the COVID-19 pandemic compared to those of the pre-pandemic period. To what extent these aspects may have played a role in the rise of azole-non-susceptible *C. parapsilosis sensu stricto* infections in our epidemiological study, and their related weighted contributions, is a complicated subject of research and needs further investigations that are for now beyond the purpose of this study.

As reported in the WHO fungal priority pathogen list (WHO FPPL) [49], healthcare professionals and stakeholders should focus their efforts on responding appropriately to this worrisome matter of public health. Interventions should aim at preventing the development of antifungal resistance through appropriate laboratory diagnosis, identification of the resistant microorganism and implementation of antifungal stewardship programs. As mentioned in the *C. parapsilosis*-specific chapter of the WHO FPPL, it is of the utmost importance to set up a systematic surveillance program of both susceptible and non-susceptible strains as well as report and define in-hospital epidemiology data such as incidence, prevalence and antifungal susceptibility patterns of *C. parapsilosis* among candidemia cases [38,49].

Future research should be aimed at investigating molecular epidemiology and performing whole-genome sequencing of *C. parapsilosis sensu stricto* isolates, to investigate the underlying resistance mechanisms.

## 5. Conclusions

We report a paradigm shift in the hospital clinical epidemiology of candidemia from a large tertiary care center. *Candida parapsilosis sensu stricto* was found to be the most prevalent isolate among *Candida* spp. recovered from blood cultures, with worrisome fluconazole- and voriconazole-non-susceptibility rates, unparalleled among European and North American geographical regions. Since *Candida parapsilosis sensu stricto* non-susceptibility to azole compounds has essentially replaced the in-hospital ecologic niche that was previously occupied by fluconazole- and voriconazole-susceptible *Candida albicans*, *Candida parapsilosis sensu stricto* and other *Candida* species, we question whether the same epidemiologic shift might be seen or is already occurring in other centers if the same epidemiological research protocol is into practice.

## Figures and Tables

**Figure 1 jof-09-01012-f001:**
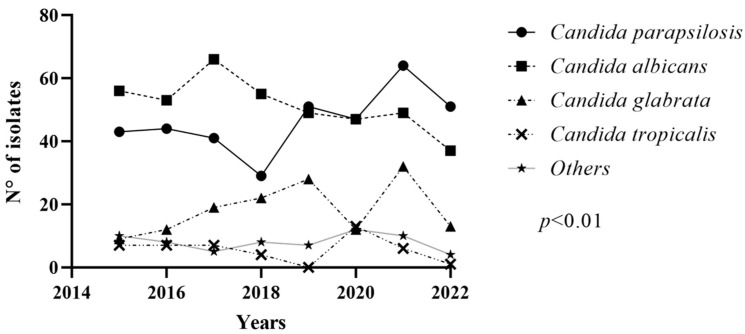
Number of *Candida* species isolated from blood cultures per year. Schematic representation of the number of *Candida* species isolated from blood cultures from January 2015 to August 2022. For each species and each affected patient, only the first positive blood culture for *Candida* spp. over a 30-day period was included in this study. BSI episodes caused by the same species were, therefore, excluded if detected within 30 days from the first positive blood culture.

**Figure 2 jof-09-01012-f002:**
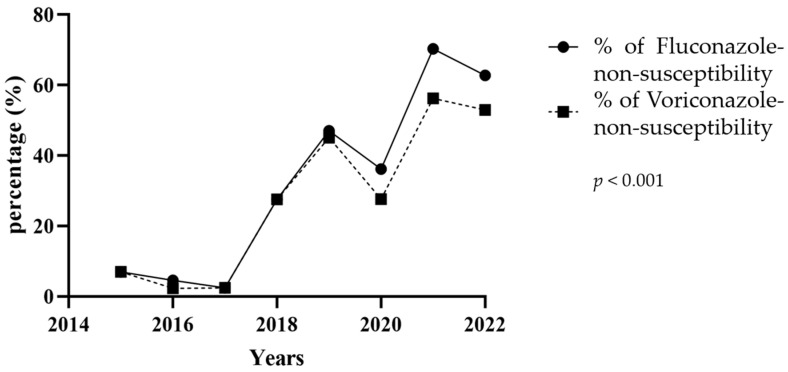
Percentage of *Candida parapsilosis sensu stricto* non-susceptible isolates. Percentage of fluconazole- and voriconazole-non-susceptible *Candida parapsilosis sensu stricto* strains isolated from blood cultures during the study period (January 2015 to August 2022). Non-susceptibility includes both intermediate and resistant MIC values.

**Table 1 jof-09-01012-t001:** Distribution and prevalence of *Candida* species isolated from blood cultures per year.

	No. of Isolates (%) Incidence (I)/10,000 Patient-Day											
	*C. parapsilosis sensu stricto*		*C. albicans*		*C. glabrata*		*C. tropicalis*		Others ***		Overall	
Year	N (%)	I	N (%)	I	N (%)	I	N (%)	I	N (%)	I	N	I
2015	44 (34.9)	1.44	56 (44.4)	1.88	9 (7.2)	0.3	7 (5.5)	0.24	10 (8)	0.34	126	4.24
2016	44 (35.5)	1.5	53 (42.7)	1.81	12 (9.7)	0.41	7 (5.6)	0.24	8 (6.5)	0.28	124	4.23
2017	41 (29.7)	1.38	66 (47.8)	2.24	19 (13.8)	0.64	7 (5.1)	0.24	5 (3.6)	0.17	138	4.68
2018	29 (24.6)	0.99	55 (46.6)	1.89	22 (18.6)	0.76	4 (3.4)	0.14	8 (6.8)	0.27	118	4.05
2019	51 (37.8)	1.67	49 (36.3)	1.6	28 (20.7)	0.914	0 (0)	0	7 (5.2)	0.23	135	4.41
2020	47 (35.9)	1.87	47 (35.9)	1.87	12 (9.2)	0.48	13 (9.9)	0.52	12 (9.2)	0.48	131	5.21
2021	64 (39.8)	2.45	49 (30.4)	1.87	32 (19.9)	1.22	6 (3.7)	0.23	10 (6.2)	0.38	161	6.16
2022 ^§^	51 (48.1)	N/A ^$^	37 (34.9)	N/A	13 (12.3)	N/A	1 (0.9)	N/A	4 (3.8)	N/A	106	N/A

* Includes: 19 *C. krusei*, 17 *C. guillermondii*, 10 *C. orthopsilosis*, 4 *C. dubliniensis*, 4 *C. metapsilosis*, 3 *C. lusitanie*, 2 *C. incospicua*, 2 *C. pelliculosa*, 1 *C. famata*, 1 *C. rugosa*, 1 *C. lipolytica*. ^§^ Data are related to the period 1 January 2022–31 August 2022. ^$^ Data regarding total patient-days for 2022 have not yet been published by Azienda Ospedaliero-Universitaria Pisana.

**Table 2 jof-09-01012-t002:** Distribution of *Candida parapsilosis sensu stricto* vs. other *Candida* spp. per year according to clinical settings.

	No. of *C. parapsilosis sensu stricto* Isolates/Other *Candida* spp. Clinical Settings			
Year	Internal Medicine Wards	Surgical Wards	Intensive Care Wards	Overall
2015	28/49	7/17	9/16	44/82
2016	27/45	9/20	8/15	44/80
2017	20/53	9/20	12/24	41/97
2018	16/46	6/17	7/26	29/89
2019	19/47	9/18	23/19	51/84
2020	22/55	9/10	16/19	47/84
2021	24/64	13/23	27/10	64/97
2022	25/38	10/6	16/11	51/55
Total	181/397	72/131	118/140	371/668

**Table 3 jof-09-01012-t003:** Distribution of *Candida parapsilosis sensu stricto* isolates non-susceptible to fluconazole and voriconazole per year according to clinical settings.

Drug Non-Susceptibility		No. of Non-Susceptible Isolates/Total No. of *C. parapsilosis sensu stricto* Isolates (%) Clinical Settings		
	Year	Internal Medicine Wards	Surgical Wards	Intensive Care Wards
Fluconazole	2015	3/28 (10.7)	0/7 (0)	0/9 (0)
	2016	0/27 (0)	1/9 (11.1)	1/8 (12.5)
	2017	1/20 (5)	0/9 (0)	0/12 (0)
	2018	1/16 (6.2)	1/6 (16.7)	6/7 (85.7)
	2019	7/19 (36.8)	3/9 (33.3)	14/23 (60.8)
	2020	7/22 (31.8)	1/9 (11.1)	9/16 (56.3)
	2021	10/24 (41.7)	10/13 (76.9)	25/27 (92.6)
	2022	11/25 (44)	7/10 (70)	14/16 (87.5)
	Total	40/181 (22.1)	23/72 (31.9)	69/118 (58.4)
Voriconazole	2015	3/28 (10.7)	0/7 (0)	0/9 (0)
	2016	0/27 (0)	0/9 (0)	1/8 (12.5)
	2017	1/20 (5)	0/9 (0)	0/12 (0)
	2018	1/16 (6.2)	1/6 (16.7)	6/7 (85.7)
	2019	6/19 (31.6)	3/9 (33.3)	14/23 (60.9)
	2020	4/22 (18.2)	1/9 (11.1)	8/16 (50)
	2021	6/24 (25)	7/13 (53.8)	23/27 (85.2)
	2022	7/25 (28)	6/10 (60)	14/16 (87.5)
	Total	28/181(15.5)	18/72 (25)	66/118 (55.9)

## Data Availability

All necessary data required to evaluate the conclusions of this study are reported in the tables in the main text. Data regarding hospital patient-days and medical activities and reports can be accessed without restriction from the Azienda Ospedaliero Universitaria Pisana publicly available repository website through the following link: https://www.ao-pisa.toscana.it/index.php?option=com_content&view=article&id=5650:relazione-sanitaria-2021&catid=264&Itemid=112 (accessed on30 September 2023). Restrictions apply to the availability of these data. For legal and ethical reasons, the study data are available upon formal and reasonable request to the corresponding author, which will be submitted for the approval of the Tuscany Area Vasta Nord Ethical Committee.

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
