# Peer review of "Paradigm Shift: Candida parapsilosis sensu stricto as the Most Prevalent Candida Species Isolated from Bloodstream Infections with Increasing Azole-Non-Susceptibility Rates: Trends from 2015–2022 Survey"

_jof, 2023, doi:10.3390/jof9101012_

Round 1

Reviewer 1 Report

In September 2023, I received a request from Journal of Fungi to review the manuscript “Paradigm shift: Candida parapsilosis as the most prevalent Candida species isolated from bloodstream infections with increasing azole-non-susceptibility rates. Trends from 2015-2022 survey”, that describes episodes of candidemia in a hospital in Italy (from 2015 to 2022) and trend to non-susceptibility to C. parapsilosis, according to the authors.
I would reject this study for 3 main reasons.
1) The Method section is not clear. Study period from 2015 to 2022. Why only 2021 has epidemiological information?
Absolute no information about yeast identification, except MALDI-TOF.
The 2 AFST (antifungal susceptibility testing) method are commercial and good for screening.
2) There is no way to confirm today the non-susceptible azoles C. parapsilosis unless the authors use Gold Standard method and mainly assays to identify mutation and molecular mechanisms for resistance.
Some paragraphs are confused and should be re-writen again.

Author Response

Reviewer#1

In September 2023, I received a request from Journal of Fungi to review the manuscript “Paradigm shift: Candida parapsilosis as the most prevalent Candida species isolated from bloodstream infections with increasing azole-non-susceptibility rates. Trends from 2015-2022 survey”, that describes episodes of candidemia in a hospital in Italy (from 2015 to 2022) and trend to non-susceptibility to C. parapsilosis, according to the authors.
I would reject this study for 3 main reasons.
1) The Method section is not clear. Study period from 2015 to 2022. Why only 2021 has epidemiological information?
Absolute no information about yeast identification, except MALDI-TOF.
The 2 AFST (antifungal susceptibility testing) method are commercial and good for screening.
2) There is no way to confirm today the non-susceptible azoles C. parapsilosis unless the authors use Gold Standard method and mainly assays to identify mutation and molecular mechanisms for resistance.
Some paragraphs are confused and should be re-writen again.

We thank Reviwer #1 for these useful and important observations. Answers to each issue raised in this section have been properly addressed specifically in the following points. Paragraphs highlighted as confusing were re-written accordingly.

 Line Manuscript information/issue Recomendation / Doubts

  1. INTRODUCTION

35 It is missing information about C. parapsilosis as a complex of species (C. parapsilosis, C. orthopsilosis, C. metapsilosis) mainly for the fact that this study recovered at 14 episodes of C. orthopsilosis and C. metapsilosis.- mention C. parapsilosis complex (include Tavanti et al. 2005 JCM and/or Tavanti et al. 2010 BMC). More information can also be find in the cited reference Tóth et al. 2019 CMR

We thank Reviewer #1 for the important suggestions and corrections. According to the first one here reported we have added this new tense to address it:

Line 79-83. Initially, genetic studies on several C. parapsilosis strains identified three genotypically different groups defined as “C. parapsilosis complex”[22]. Subsequently, these groups were redefined as new species by Tavanti et al. under the name of C. orthopsilosis, C. metapsilosis and C. parapsilosis sensu stricto [23]

Also two new references were added:

22            Roy, B.; Meyer, S.A. Confirmation of the Distinct Genotype Groups within the Form Species Candida Parapsilosis. J Clin Microbiol 1998, 36, 216–218, doi:10.1128/JCM.36.1.216-218.1998.

  1. Tavanti, A.; Davidson, A.D.; Gow, N.A.R.; Maiden, M.C.J.; Odds, F.C. Candida Orthopsilosis and Candida Metapsilosis Spp. Nov. To Replace Candida Parapsilosis Groups II and III. Journal of Clinical Microbiology 2005, doi:10.1128/jcm.43.1.284-292.2005.

45-49 It is missing candidemia species distribution from some continents. Beyond Guinea, 2014, in Asia Tan et al. 2016 Med Mycol show it more precisely in several countries. In Latin America, an interesting review from da Matta et al. 2017 JoF Include the references Tan et al. 2016 Med Mycol and da Matta et al. 2017 JoF

Line 49-60. In the study by Tan et al [9], published in 2016, on species distribution and antifungal susceptibility profile of invasive Candida spp bloodstream infections, C. albicans was found to be the most frequently isolated species in the Asia-pacific region, followed by C. tropicalis and thirdly C. parapsilosis with prevalence of 35.9%, 30.7% and 15.7% respectively, along with susceptibility rates for fluconazole of 99.7%, 75.8% and 94.8%. In another study, da Matta and colleagues investigated retrospectively the epidemiology of invasive BSIs caused by Candida spp. in Central and Latin America [10]. Authors evaluated 40 studies from 2007 to 2016 and found that C. albicans was the first species causing candidemia, followed by C. parapsilosis and then C. tropicalis [10]. Trends in fluconazole resistant strains showed an increase over time for all the three above mentioned species from 0.4-1.2% for C. albicans, from 0.5-2.3% for C. tropicalis and from 0-2.6% for C. parapsilosis [10].

The following references have been added:

  1. Tan, T.Y.; Hsu, L.Y.; Alejandria, M.M.; Chaiwarith, R.; Chinniah, T.; Chayakulkeeree, M.; Choudhury, S.; Chen, Y.H.; Shin, J.H.; Kiratisin, P.; et al. Antifungal Susceptibility of Invasive Candida Bloodstream Isolates from the Asia-Pacific Region. Med Mycol 2016, 54, 471–477, doi:10.1093/mmy/myv114.
  2. da Matta, D.A.; Souza, A.C.R.; Colombo, A.L. Revisiting Species Distribution and Antifungal Susceptibility of Candida Bloodstream Isolates from Latin American Medical Centers. J Fungi (Basel) 2017, 3, 24, doi:10.3390/jof3020024.

57-59 “Data from the SENTRY surveillance program reported by Pfaller and colleagues in 2019 [8] showed that the second most common isolated Candida species was Candida glabrata with the exception of Latin America.” SENTRY is a really important surveillance for species distribution, although there is no information about candidemia incidence. For it, da Matta et al. 2017 JoF present an analysis of several studies and countries, including Caribbean. Include Matta et al. 2017 JoF

Line 71-74 As previously mentioned, the work of da Matta and colleagues evaluated trends in epidemiology of invasive candida bloodstream infections in central and Latin America highlighting C. parapsilosis as the second most frequently fungal isolate causing candidemia [10].

61 Regarding outbreak of C. parapsilosis. It is missing a study describing one of the first outbreak of these complex of species in candidemia in South America (Pinhati et al. 2016 BMC) Include Pinhati et al 2016 BMC

Line 76. […] Latin America [20] and Turkey […].

The following reference has been added:

  1. Pinhati, H.M.S.; Casulari, L.A.; Souza, A.C.R.; Siqueira, R.A.; Damasceno, C.M.G.; Colombo, A.L. Outbreak of Candidemia Caused by Fluconazole Resistant Candida Parapsilosis Strains in an Intensive Care Unit. BMC Infect Dis 2016, 16, 433, doi:10.1186/s12879-016-1767-9.

90-100 This kind of subjects and recommendation is not usually find in the INTRODUCTION. This information should be in the DISCUSSION.

The following part has been moved from the introduction to the discussion section:

Line 468-477. As reported in the document WHO fungal priority pathogen list (WHO FPPL) [49] health-care professionals and stakeholders should focus all their efforts on responding appropriately to this worrisome matter of public health. Interventions should aim at preventing the development of antifungal resistance through appropriate laboratory diagnosis and identification of the resistant microorganism and implementation of antifungal stewardship programs. As mentioned in the WHO FPPL in the C. parapsilosis specific chapter it is of the outmost importance to set up a systematic surveillance program of both susceptible and non-susceptible strains as well as reporting and defining in-hospital epidemiology data such as incidence, prevalence and antifungal susceptibility patterns of C. parapsilosis among candidemia cases [38,49].

  1. MATERIALS AND METHODS

109-115 This text is really confusing, if the study period is from 2015 to 2022, why the authors provided so much information for 2021? Re-write it in a clear and objective way

Line 139-149. In order to depict the clinical context in which this study has been performed, it is important to mention that according to the latest report provided by the healthcare management in 2021, Pisa University Hospital hosted 1108 beds and had 49045 admissions in the same year. Surgical wards total admissions and beds were 21279 and 444. Internal medicine total admissions and beds in 2021 were 20268 and 439 respectively. ICU total beds were 108. In addition, a total of 262 surgical transplantation procedures were performed as follows: 140 liver, 81 bone marrow, 33 kidney and 4 pancreas. Data regarding hospital admissions, distribution of beds in relation to medical activities and specialties as well as surgical transplantation procedures were in line with those reported in previous years with the exception of 2020 when a reduction of the number of admissions was observed.

130-131 The authors used MALDI-TOF for yeasts identification. Which system, Bruker, BioMerieux, other? This is a really important information, including the database.

Line 150-151. (MALDI Biotyper Compass® Bruker Daltonics GmbH & Co. KG Bremen Germany Fahrenheitstr. 4 – 28359 Bremen).

131-173 That´s a really confusing statement. Not only the English grammar but also the methods. Re-write it in a clear and objective way. In summary it was 2 commercial colorimetric methods.

Line 164-169. Two commercial colorimetric methods were used to assess antifungal susceptibility. From January 2015 to April 2020 the antifungal susceptibility test used was Sensititre YeastOne Y010© (Thermo Fisher Diagnostics B.V., Lansmeer, The Netherlands). From May 2020 to August 31st 2022 the applied method was Merlin MICRONAUT-AM Antifungal Agents MIC© (Bruker Daltonics GmbH & Co. KG, Germany).

151-152 “incubated at 37oC” Check the write temperature. This is really basic to Candida in vitro assays.

Line 183-184. […] plates were then incubated at 35-37°C

We decided to maintain 37°C according to Merlin Micronaut manufacturers’ instructions https://www.merlin-diagnostika.de/fileadmin/mediapool/downloads/Produkte/Micronaut_Spezialplatten/1875129_MICRONAUT-AM_03-2020_eBook.pdf )

164-167 Information about EVC. How did the use ECV in this analysis? Review it, it is not clear how did the authors used this tool.

Line 197-201. For amphotericin B and posaconazole susceptibility interpretation, MIC values were compared to Epidemiological Cutoff Values (ECV). MIC values for C. parapsilosis above the ECV (amphotericin B >1 µg/mL, posaconazole > 0.25 µg/mL) were retested with E-test strip. In cases of concordant results the yeast isolate would have been classified as “non-wild-type” [42].

181-185 I have no idea what does the authors mean with this statement Review it.

According to reviewer’s suggestions we decided to erase the following part:

We decided to rephrase it as follows:

Line 225-227. In all statistical analyses C. parapsilosis sensu stricto isolates with intermediate or resistant phenotypes for fluconazole and/or voriconazole were regrouped as “non-susceptible” for the corresponding molecule.

  1. RESULTS

207-211 The sum of the percentage should be 100%, not 99.8% Calculate it again.

Line 245-146. Overall species distribution from 2015 to 2022 showed 412 (39.7%) cases of C. albicans, 371 (35.7%) C. parapsilosis, 147 (14.1%) C. glabrata, 45 (4.3%) C. tropicalis, 64 (6.2%

220 Table 1. It is also not clear why the authors didn’t make the analysis for the complex? (C. parapsilosis, C. orthopsilosis and C. metapsilosis) Make the analysis for the C. parapsilosis complex

We thank the reviewer for this comment as it outlines a possible source of misunderstanding for the reader. Since the outbreak retrieved in our hospital and the associated increase in azole non-susceptibility pattern was caused only by C. parapsilosis sensu stricto we decided to make the analysis only for this species rather than for the entire complex. Plus, pooled prevalence of C. orthopsilosis and C. methapsilosis among all candidemia episodes was 1.3% and fluconazole and voriconazole MIC values were below the ECVs for the corresponding molecules. Therefore, based on the reviewer’s comments, we decided to add the following tense to clarify our decision in the Discussion section:

Line 249-252. Since the pooled prevalence of C. orthopsilosis and C. metapsilosis among all candidemia episodes was 1.3%, we decided to perform analysis only for C. parapsilosis sensu stricto rather than for the entire complex.

According to this the following specification of Candida parapsilosis sensu stricto were made throughout the text.

250 Table 1. It seems there is no difference between general Candida species distribution vs C. parapsilosis for the clinical settings distribution. Useful to show the general Candida species distribution, or create something to compare general vs C. parapsilosi

We thank the reviewer for this useful suggestion. In order to compare general Candida species distribution per clinical setting vs Candida parapsilosis we decided to modify Table 2 instead of Table 1 as follows since it fitted better the reviewer’s request:

Table 2. Distribution of Candida parapsilosis sensu stricto vs Other Candida spp per year according to Clinical Settings.

No. of C. parapsilosis sensu stricto isolates/Other Candida spp.

Clinical Settings

Year

Internal Medicine Wards

Surgical Wards

Intensive Care Wards

Overall

2015

28/49

7/17

9/16

44/82

2016

27/45

9/20

8/15

44/80

2017

20/53

9/20

12/24

41/97

2018

16/46

6/17

7/26

29/89

2019

19/47

9/18

23/19

51/84

2020

22/55

9/10

16/19

47/84

2021

24/64

13/23

27/10

64/97

2022

25/38

10/6

16/11

51/55

Total

181/397

72/131

118/140

371/668

Table 2. Non-susceptible azoles C. parapsilosis strains should be confirmed by gold standard methods (CLSI or EUCAST) and molecular resistance assays. Apply molecular assays to confirm that C. parapsilosis is non-susceptible to azoles.

We agree with Reviewer’s suggestion and comments, unfortunately no molecular method was and still is available in our hospital to detect antifungal resistance in yeasts. However as reported also in the response to the editor we have provided adjunctive data that were not previously included in the study regarding our E-test strip test performed as a secondary test to confirm the resistant phenotype.

Line 277-279. All isolates presenting intermediate and resistant phenotypes to fluconazole and voriconazole were retested with E-test strip which confirmed the non-susceptible phenotype.

  1. DISCUSSION

314 It was nothing mentioned about the impact of COVID19 in the candidemia episodes. Please read Ramos-Martínez et al. 2022 JoF as an example

Mention the impact of COVID19 in candidemia episodes and insert Ramos-Martínez et al. 2022 JoF as an example

Line 460-464. Indeed, in an interesting study conducted by Ramos-Martinez et al. [46] COVID-19 pandemic greatly affected the incidence of candidemia, especially in the case of C. parapsilosis. In their work, authors highlighted an increase in both incidence rates and associated fluconazole resistance of C. parapsilosis bloodstream infections during the COVID-19 pandemic compared to those of pre-pandemic period.

328-333 “The results reported… To our knowledge a similar scenario has not yet been described in Europe [24]”. The method was used to classify non-susceptible is not accurated. The authors cited some studies, for example, Trevijano-Contador et al. 2022 Open Forum Infect Dis. In that study it was confirmed the C. parapsilosis resistant using molecular biology methods. The authors should considerate some other variables to confirm it. Confirm the non-susceptilble strains. The impact of COVID19 in the current hospital. Be extremely careful before present these results as a trend in Europe

According to Reviewer’s suggestions we decided to erase the following tense:

Line 396. To our knowledge a similar scenario has not yet been described in Europe [29]

and decided to add the following correction:

Line 388-392. Our results should be interpreted with caution. Due to the lack of a molecular method in our hospital for the detection of underlying resistance mechanisms in Candida spp., we were not able to perform any adjunctive confirmatory test other than E-test strip, which still follows a phenotypic approach. Regardless, whole genome sequencing and adjunctive molecular tests will be the subject of future investigations and research.

Reviewer 2 Report

In this manuscript the authors describes the prevalence of C. parapsilosis and other blood stream infections caused by pathogenic yeast. The study also finds that there is an increase in the incidence of C. parapsilosis and observed azole resistance in this pathogenic fungi, particularly post 2018. Over all this is an interesting article and suitable for publication in Fungi. However, I suggest the authors to address the following points. 

1) The method described for the identification and culturing of the isolated fungi only mention samples collected from 2020-2022. It is not clear, how did the authors isolated and characterized prior to this period. The authors should clearly describe the details in the revised version of the manuscript. Also, if the authors received only the cultures ( isolated strains), they should make sure that culture/isolation  conditions during this period did not had any biasness toward any other species. 

2) The age of the patients vary from 1-115, it would be ideal show if there was any age and or sex  association with the increased C. parapsilosis incidence.  

Author Response

Reviewer #2

In this manuscript the authors describes the prevalence of C. parapsilosis and other blood stream infections caused by pathogenic yeast. The study also finds that there is an increase in the incidence of C. parapsilosis and observed azole resistance in this pathogenic fungi, particularly post 2018. Over all this is an interesting article and suitable for publication in Fungi. However, I suggest the authors to address the following points. 

1) The method described for the identification and culturing of the isolated fungi only mention samples collected from 2020-2022. It is not clear, how did the authors isolated and characterized prior to this period. The authors should clearly describe the details in the revised version of the manuscript. Also, if the authors received only the cultures ( isolated strains), they should make sure that culture/isolation  conditions during this period did not had any biasness toward any other species. 

We thank the Reviewer for these useful observations and corrections. Since text was not clear we decided to revise and change line147-150 accordingly:

Line 146-150 were erased.

Line150 -160. For the entire study period (2015-2022) yeast isolates recovered from positive blood cultures were identified with MALDI-TOF MS (MALDI Biotyper Compass® Bruker Daltonics GmbH & Co. KG Bremen Germany). The microbiology laboratory supervised and handled the complete management of blood cultures taken from patients with suspected sepsis. The diagnostic workflow of BSIs in our hospital is as follows: Blood cultures (BACTEC Plus/F Aerobic, BACTEC Plus/F Anaerobic, BACTEC Peds Plus/F) from patients with suspected sepsis were sent to the microbiology laboratory where they were incubated in the BD BACTEC FX system (Becton Dickinson and Company; Milano, Italia) at 35 +/− 1.5 °C for up to 7 days. All positive blood cultures then underwent Gram staining and were plated also onto Sabouraud agar plates amongst other solid media. Plates were then incubated for 24 h at 35-37°C. Yeast colonies were then identified using MALDI-TOF MS.

2) The age of the patients vary from 1-115, it would be ideal show if there was any age and or sex  association with the increased C. parapsilosis incidence.  

We thank the reviewer for these useful suggestions. Age and sex were not associated with the increase in C. parapsilosis incidence therefore we have added the following statement:

Line 301-302. No association was found between age and sex were not with the increased prevalence of C. parapsilosis sensu stricto

Reviewer 3 Report

In this study by Franconi et al entitled “Paradigm shift: Candida parapsilosis as the most prevalent Candida species isolated from bloodstream infections with increasing azole-non-susceptibility rates. Trends from 2015-2022 survey.” The study was a retrospective review of patients with candidemia in their centers from 2015 to 2022. The study found a trend of increased C. parapsilosis compared to other Candida species, with concerning azole resistance. This is an important observation as it highlights the rise of non-albicans Candida species associated with bloodstream infections. Moreover, the study emphasizes the regional differences in the prevalence of non-albicans Candida species, with C. parapsilosis being more common in some regions compared to other Candida species such as C. glabrata. The study also found that cases of C. parapsilosis that are resistant to azoles are on the rise.

I have a few comments to the authors regarding the study. 

Line 179, in the data for candidemia, the link is broken. Can the authors update the link and indicate the date the data were accessed?

Table 2 does not show the result of different wards and the rise of C. parapsilosis. One suggestion is to make the denominator all Candida spp, instead of C. parapsilosis. 

It is very informative to have both fluconazole and voriconazole susceptibility testing, but in IDSA guidelines, it is not recommended to use other azoles if Candida spp. is resistant to fluconazole, besides C. krusei. What is the authors’ center’s approach? Do they use voriconazole in cases of fluconazole-resistant C. parapsilosis?

The results of C. parapsilosis increase are very salient, and it is interesting to see ICU patients are risk factors. Is this secondary to line-related infections? Do the authors have more granular information regarding the source of the candidemia, e.g. line, devices, intraabdominal..etc?

Similarly, do the authors have information regarding the type of prior antifungal exposure in these settings? 

Lastly, is the C. parapsilosis azole-resistant matching or higher than the other Candida spp.?

In the discussion, the authors mentioned that they could not perform molecular analysis. Does that mean they are thinking of studying the in-hospital spread of this strain? There is concern that such a strain of C. parapsilosis has spread among patients from infection control as reported before in a Spanish cohort; there should be a discussion regarding this issue. Also, if the information is available, was there any cluster of patients' rooms/wards where the isolates were higher than others?

References

Díaz-García J, Gómez A, Alcalá L, et al. Evidence of Fluconazole-Resistant Candida parapsilosis Genotypes Spreading across Hospitals Located in Madrid, Spain and Harboring the Y132F ERG11p Substitution. Antimicrob Agents Chemother. 2022 Aug 16;66(8):e0071022. doi: 10.1128/aac.00710-22. Epub 2022 Jul 19. PMID: 35852369; PMCID: PMC9380585.

Author Response

Reviewer #3

In this study by Franconi et al entitled “Paradigm shift: Candida parapsilosis as the most prevalent Candida species isolated from bloodstream infections with increasing azole-non-susceptibility rates. Trends from 2015-2022 survey.” The study was a retrospective review of patients with candidemia in their centers from 2015 to 2022. The study found a trend of increased C. parapsilosis compared to other Candida species, with concerning azole resistance. This is an important observation as it highlights the rise of non-albicans Candida species associated with bloodstream infections. Moreover, the study emphasizes the regional differences in the prevalence of non-albicans Candida species, with C. parapsilosis being more common in some regions compared to other Candida species such as C. glabrata. The study also found that cases of C. parapsilosis that are resistant to azoles are on the rise.

I have a few comments to the authors regarding the study. 

Line 179, in the data for candidemia, the link is broken. Can the authors update the link and indicate the date the data were accessed?

We would like to thank the reviewer for these important observations and corrections

The following link has been upgraded and integrated with a reference where last date of access could be seen at the end of the citation:

Line 213-214:

https://www.ospedaledipisa.it/index.php?option=com_content&view=article&id=5650:relazione-sanitaria-2021&Itemid=112

The following reference has been added:

  1. Relazione Sanitaria 2021 Available online: https://www.ospedaledipisa.it/index.php?option=com_content&view=article&id=5650:relazione-sanitaria-2021&Itemid=112 (accessed on 30 September 2023).

Table 2 does not show the result of different wards and the rise of C. parapsilosis. One suggestion is to make the denominator all Candida spp, instead of C. parapsilosis. 

We thank the Reviewer for this suggestion. According to a previous revision proposed by Reviewer #1 we have modified table 2 comparing the number of C. parapsilosis sensu stricto vs Other Candida spp. isolated from blood cultures for each year of the study period and regrouped them according to different wards. Herein we report the modifications previously mentioned which we believe to be also suitable for an answer to this comment.

Line 292.

Table 2. Distribution of Candida parapsilosis sensu stricto vs Other Candida spp per year according to Clinical Settings.

No. of C. parapsilosis sensu stricto isolates/Other Candida spp.

Clinical Settings

Year

Internal Medicine Wards

Surgical Wards

Intensive Care Wards

Overall

2015

28/49

7/17

9/16

44/82

2016

27/45

9/20

8/15

44/80

2017

20/53

9/20

12/24

41/97

2018

16/46

6/17

7/26

29/89

2019

19/47

9/18

23/19

51/84

2020

22/55

9/10

16/19

47/84

2021

24/64

13/23

27/10

64/97

2022

25/38

10/6

16/11

51/55

Total

181/397

72/131

118/140

371/668

It is very informative to have both fluconazole and voriconazole susceptibility testing, but in IDSA guidelines, it is not recommended to use other azoles if Candida spp. is resistant to fluconazole, besides C. krusei. What is the authors’ center’s approach? Do they use voriconazole in cases of fluconazole-resistant C. parapsilosis?

We thank the reviewer for these useful suggestions. We believed that most of the issues and clarifications appointed by the Reviewer should be elucidated in the Discussion section.

Line 422-425. According to international guidelines [6] and in-hospital procedures and protocols, it is not recommended to use other azoles if Candida spp. is resistant to fluconazole, but echinocandins. Since the azole non-susceptibility detected by vitro test affected also voriconazole we decided to include and report this molecule in all analyses. 

The results of C. parapsilosis increase are very salient, and it is interesting to see ICU patients are risk factors. Is this secondary to line-related infections? Do the authors have more granular information regarding the source of the candidemia, e.g. line, devices, intraabdominal..etc?

Similarly, do the authors have information regarding the type of prior antifungal exposure in these settings? 

Line 393-397. A second study limitation lies within the retrospective nature of the work presented. Unfortunately, according to the kind of information stored in the database, authors were unable to gain more granular information regarding both source of candidemia and prior exposure to antifungals. Evaluating these parameters along with survival rates and treatment outcomes with prospective studies will certainly be the subject of future research.

Lastly, is the C. parapsilosis azole-resistant matching or higher than the other Candida spp.?

This comment has been appointed in the Results section as follows:

Line 314-317. In 2021 and 2022, the number of candidemia episode of azole non-susceptible C. parapsilosis sensu stricto recovered in ICUs was higher than the global number of other Candida spp. isolates recovered from blood cultures in the same clinical ward.

In the discussion, the authors mentioned that they could not perform molecular analysis. Does that mean they are thinking of studying the in-hospital spread of this strain? There is concern that such a strain of C. parapsilosis has spread among patients from infection control as reported before in a Spanish cohort; there should be a discussion regarding this issue. Also, if the information is available, was there any cluster of patients' rooms/wards where the isolates were higher than others?

Line 430-439. Studying the in-hospital spread of this/these strain/s will be of the outmost importance for future research and infection control purposes, as such strain/s might have spread among patients within the same room/ward, as reported by a Spanish study conducted by Díaz-García et al. [47]. Defining clinical and molecular epidemiology of a similar strain has been successfully conducted by these authors, where spreading of fluconazole resistant C. parapsilosis was observed and investigated across different hospitals in Madrid [47]. However, we were unable to retrieve isolates prior to 2021, when the increase in azole non-susceptibility already took place as well as information regarding cluster of patients’ rooms/wards where recovery of azole non-susceptible C. parapsilosis sensu stricto was higher than others. 

References

Díaz-García J, Gómez A, Alcalá L, et al. Evidence of Fluconazole-Resistant Candida parapsilosis Genotypes Spreading across Hospitals Located in Madrid, Spain and Harboring the Y132F ERG11p Substitution. Antimicrob Agents Chemother. 2022 Aug 16;66(8):e0071022. doi: 10.1128/aac.00710-22. Epub 2022 Jul 19. PMID: 35852369; PMCID: PMC9380585.

This reference has been added to the discussion as follows:

  1. Díaz-García, J.; Gómez, A.; Alcalá, L.; Reigadas, E.; Sánchez-Carrillo, C.; Pérez-Ayala, A.; Gómez-García de la Pedrosa, E.; González-Romo, F.; Merino-Amador, P.; Cuétara, M.S.; et al. Evidence of Fluconazole-Resistant Candida Parapsilosis Genotypes Spreading across Hospitals Located in Madrid, Spain and Harboring the Y132F ERG11p Substitution. Antimicrob Agents Chemother 2022, 66, e0071022, doi:10.1128/aac.00710-22.

Round 2

Reviewer 1 Report

Dear all

I received the article jof-2624117 "Paradigm shift: Candida parapsilosis as the most prevalent Candida species isolated from bloodstream infections with increasing azole-non-susceptibility rates. Trends from 2015-2022 survey". 

I had suggested the “rejection” of the previous version due to the lack of a clear methodology for confirming resistance. The addition of the e-test "metthod"to this and the caution in interpreting the results of a tendency towards increased non-susceptibility to azotics made the study design more objective and the conclusion more moderate.

The second strong point was the presentation of C. parapsilosis as a complex based on the study by Tavanti et al (2005).

Finally, other references added made this subject more comprehensive and interesting for researchers from different regions of the planet.

Congratulations!